# Effects of Risk Perception of Pests and Diseases on Tea Famers’ Green Control Techniques Adoption

**DOI:** 10.3390/ijerph19148465

**Published:** 2022-07-11

**Authors:** Hai Hu, Andi Cao, Si Chen, Houjian Li

**Affiliations:** 1School of Marxism, Sichuan Agricultural University, Chengdu 611130, China; 13876@sicau.edu.cn; 2Southwestern Poverty Reduction and Development Research Center, Sichuan Agricultural University, Chengdu 611130, China; 3College of Economics, Sichuan Agricultural University, Chengdu 611130, China; caoandi@stu.sicau.edu.cn (A.C.); chensi@stu.sicau.edu.cn (S.C.)

**Keywords:** risk perception, green control techniques, tea planting, Unified Theory of Acceptance and Use of Techniques

## Abstract

Green control techniques support the concept of green plant protection, advocate for the safe and reasonable use of pesticides, and finally achieve the goal of controlling pests and diseases and protecting the environment. The purpose of this study is to explore the effect of risk perception of pests and diseases on farmers’ usage intention of green control techniques. Based on 747 samples of tea farmers in Sichuan province, China, introducing the Unified Theory of Acceptance and Use of Technology (UTAUT) framework and using the Partial Least Square–Structural Equation Modeling (PLS-SEM) approach, this paper found that risk perception has a negative impact on behavioral intention. Performance expectancy, effort expectancy, and social influence can positively affect behavioral intention, and facilitating conditions can also positively influence usage behavior. Moreover, the mediating analysis indicated that the higher the risk perception is, the less performance expectancy of green control techniques and the weaker the behavioral intention. Meanwhile, risk perception also plays a mediating effect on the relationship between effort expectancy and behavioral intention. This study could help to provide references for policymaking to improve the adoption of green control techniques.

## 1. Introduction

China is a major tea-producing country, and tea has a long history in China [1,2]. As an important cash crop, tea is the basis for many rural farmers to sustain their livelihoods. Therefore, the development of the tea industry is crucial for tea farmers. However, tea plants are susceptible to a variety of pests such as nematodes, termites, cockchafer grub, stem pests, stem borer, leaf pests, and all mite species. If these pests and diseases are not effectively controlled, they could cause 10% to 20% of yield losses [3,4]. Since tea farmers are risk-averse, they have to apply pesticides to rapidly control pests and diseases for reducing production losses [5,6]. However, excessive usage of pesticides for reducing losses has created a serious problem of agricultural pollution. Some of the pesticide residues enter the ecosystem through the atmosphere, soil, and groundwater, causing incalculable damage to the ecosystem [7], and the pesticide residues on tea leaves can affect human health when brewed [4].

To curb the excessive use of chemical pesticides and ensure food quality and safety and ecological sustainability, the Chinese government emphasizes the promotion and application of green control techniques. Similar to integrated pest management (IPM) [8], green control techniques enable the effective control of crop pests and diseases by reducing chemical pesticide use and adopting techniques such as agricultural control, physical control, biological control, ecological regulation, and the scientific pesticide use to ensure the agricultural production safety, quality safety, and agroecological environment safety, as well as to improve farmers’ welfare. However, data from the Ministry of Agriculture and Rural Development showed that by the end of 2021, the adoption rate of green control techniques in China still had not reached 50%. The reasons for the low adoption rate can be attributed to several aspects. First, although the adoption of green control techniques can bring positive environmental effects, their effectiveness in controlling pests and diseases is not immediate compared to the traditional means of applying chemical pesticides [6]. Second, the adoption of green control techniques requires farmers to learn the knowledge and operational skills and even to shoulder the high cost of technology transfer and usage risk [9]. Finally, farmers have insufficient knowledge about pesticide application and weak awareness of ecological environment protection [5].

Moreover, many studies have shown that risk perception is one of the main factors influencing farmers’ adoption of agricultural techniques. For example, Liu and Huang [10] found that risk-averse farmers tended to apply more pesticides. Fernandez-Cornejo et al. [11] pointed out that farmers who adopted the IPM tended to be less risk-averse compared to non-adopters. Li et al. [12] found that risk perception positively mediated the relationship between farmer differences and soil and water conservation techniques. Pingali et al. [13] argued that the greater the risk farmers perceived, the more they intend to adopt pest control. The high risk of pests and diseases could lead to crop yield reduction, thus affecting their choice of control techniques. To our knowledge, although studies have discussed the impact of risk perception on pesticide use and sustainable technique adoption in agriculture, there is little literature examining the impact of risk perception of pests and diseases on the usage behavior and behavioral intentions of green control techniques. In recent years, many countries have emphasized sustainable and green development in agriculture. The development of green agriculture implies the transformation of agricultural production methods, and the transformation of control methods is also a key to it. It is well known that the promotion and application of green control techniques have a positive effect on the improvement of the ecological environment and agricultural product quality. Therefore, it is important to explore the influence of risk perception of pests and diseases on the adoption of green control techniques by tea farmers for the development of policies related to the promotion of green agricultural techniques. As alluded above, this paper involves the risk perception of pests and diseases in the Unified Theory of Acceptance and Use of Technology (UTAUT) framework, using the micro-data of tea farmers from Sichuan province, China, attempting to explore the impact and mechanism of risk perception on tea farmers’ adoption of green control techniques. We found that risk perception of pests and diseases has a negative effect on the behavioral intention of green control techniques. Additionally, performance expectancy, effort expectancy, and social influence have positive effects on behavioral intention, and facilitating conditions could positively affect usage behavior. Furthermore, performance expectancy can mediate the influence of risk perception on behavioral intention, and risk perception can mediate the effect of effort expectancy on behavioral intention.

This paper has some contributions. First, although the UTAUT has been used to study farmers’ adoption of electronic commerce, communication technologies, Internet of Things technologies, and project payments [14,15,16,17,18,19,20], they have been rarely applied in the context of green agricultural techniques. In this paper, we introduce risk perception of pests and diseases into the original UTAUT framework in an attempt to explore the effect of risk perception on farmers’ behavioral intention and usage behavior of green control techniques and contribute to the existing literature. Second, the data used in this paper are more representative because Sichuan is a major tea-producing province [4], and sufficient samples can also improve the estimation results under the structural equation model (SEM). Third, we also identified mediating effects of risk perception and performance expectancy. On the one hand, risk perception plays a mediating role between effort expectancy and behavioral intention. On the other hand, performance expectancy also mediates the relationship between risk perception and behavioral intention. These results are new findings for relevant studies and help us to understand the impact of risk perception of pests and diseases on the adoption of green control techniques at a deeper level. Fourth, our findings can provide policy implications for the government in making policies for the promotion of green control techniques as well as green agriculture development.

The remainder of this paper is shown as follows: Section 2 explains the theory and hypotheses; Section 3 introduces the questionnaire and data source; Section 4 and Section 5 report and discuss the results, respectively; Section 6 states the conclusions, limitations, and future research.

## 2. Theory and Hypotheses

### 2.1. Unified Theory of Acceptance and Use of Technology

The Unified Theory of Acceptance and Use of Technology (UTAUT) is often used to determine the predictors affecting technology acceptance. This model has been proven to have good predictability [21]. There are many factors related to technology adoption that can affect usage behavior. It involves four key determinants: performance expectancy, effort expectancy, social influence, and facilitating conditions. Behavioral intention is determined by three factors: performance expectation, effect expectation, and social influence. In addition, facilitating conditions affect usage behavior. The main model [22] is rooted in the traditional technology acceptance model (TAM) [23]. Based on the conceptual and empirical similarity of the eight classical models [19], they are integrated into a single model, making the UTAUT an important model for studying the adoption of technology [24,25]. In agriculture, the UTAUT has been used to study farmers’ intentions and behavior in using technologies such as Internet of Things technologies [17,18], electronic commerce [15,20], communication technologies [14,16], smartphone [26], and energy project payments [19]. However, to our knowledge, the UTAUT has rarely been applied in the field of green agricultural techniques, so this area is worth exploring. Therefore, this paper contributes to the literature in this area by using the UTAUT to study the intention and behavior of Chinese tea farmers in adopting green control techniques. More importantly, this paper introduces the risk perception of pests and diseases based on the original UTAUT to analyze the effect of risk perception on the adoption of green control techniques. Figure 1 shows the conceptual framework.

### 2.2. Behavioral Intention and Usage Behavior

Behavioral intention means the intention degree of using and applying new techniques, which is also defined as the probability for someone to make a conscious plan to achieve future certain behaviors [27]. Usage behavior refers to the actual degree of users’ technique adoption [22]. In many behavioral theoretical models, behavioral intention is a good factor for behavioral prediction [23,28,29]. The increasing studies confirmed that when people have a stronger intention to use technology, they are more likely to use it [17,19,22,24]). Thus, this study puts forward a hypothesis as follows:

**Hypothesis** **1** **(H1).**
*Behavioral intention has a positive effect on farmers’ usage behavior of green control techniques.*


### 2.3. Risk Perception

Risk perception is defined as the perception of adverse consequences caused by an individual engaging in a specific activity (such as using a specific technique) under uncertain conditions [30,31]. It is regarded as an important factor in predicting human behavior. Some scholars introduced the variable of risk perception into the UTAUT framework and discussed the effect. For instance, Khatimah [32] added the variable of risk perception into the UTAUT model, indicating that risk perception would have an impact on people’s behavioral intentions. Chen [33] studied the potential application of mobile payment and found that the risk perception of people is the direct reason why they have the intention to use mobile phones.

Pest disease is considered to be the one of biggest risks faced by farmers because it will have a decisive impact on their output and income. To control pests and diseases and reduce yield losses, they will apply pesticides. Pingali [34] studied the rice production behavior of farmers in the Philippines and found that farmers often apply excessive pesticides to avoid the risk of pests and diseases. Liu and Huang [35] also indicated that farmers with higher risk perception would choose to apply more pesticides in planting. However, such behavior would cause great damage to the environment. That is, in the face of greater risks, farmers’ lower awareness of environmental protection may be one of the reasons for the low intention to adopt green control techniques. Additionally, the research of Chu and Li [36] showed that the degree of farmers’ risk preference also affects their behavior in adopting green control techniques. Preferred risk farmers can accept the uncertainty of adopting new techniques and have more intention to apply them. Therefore, this study adds the variable of risk perception of pests and diseases into the UTAUT framework to understand farmers’ techniques adoption behavior. The hypothesis is proposed as follows:

**Hypothesis** **2** **(H2).**
*Risk perception has a negative effect on the behavioral intention of green control techniques.*


### 2.4. Performance Expectancy

Performance expectancy is the degree to which farmers believe that the use of green control technologies can help them obtain benefits [22]. Performance expectation is the one of strongest predictors of behavior intention of techniques adoption. It is important in predicting farmers’ adoption of techniques [37,38]. Many studies confirmed that performance expectancy can positively influence people’s technology adoption [39,40,41,42,43,44]. For example, Hans et al. [45] showed that if techniques can bring higher efficiency, the intention of farmers to adopt it will be stronger. Liu and Wang [10] found that the more useful farmers feel when using techniques, the stronger their behavioral intention.

Moreover, farmers generally have limited knowledge of green control techniques, and they are uncertain about their performance. For instance, Martins et al. [46] indicated that when people perceive a high risk, they may think the convenience brought by Internet banking is not so strong, so their behavioral intention is low. Tai and Ku [47] believed that when stock investors trade stocks through mobile devices, they are required to provide some private information, which may be used by some hackers for improper operations. Therefore, when investors are aware of the risks, they may decide to give up the potential benefits of using mobile stock trading. Most of the current studies have shown that when people perceive a high risk of using such a system or technology, they feel that the benefits of the system will decline. In this study, if farmers perceive a higher risk of pests and diseases, they would not believe that green control techniques produce significant performance effects, thereby leading to a lower intention to use them. Therefore, this study presents two hypotheses as follows:

**Hypothesis** **3** **(H3).**
*Performance expectancy has a positive effect on behavioral intention to use green control techniques.*


**Hypothesis** **3M** **(H3M).**
*Performance expectancy of green control techniques mediates the effect of risk perception on the behavioral intention of green control techniques.*


### 2.5. Effort Expectancy

Effort expectancy is the perceived ease associated with the use of techniques [22]. It comes from the TAM model and is adapted from “perceived ease of use”. Some scholars found that effort expectancy positively impacts behavioral intention [43,48,49]. Davis et al. [50] and Sorebo and Eikebrokk [51] found that farmers’ intentions would be higher when they think that the difficulty of applying the techniques is small. Paul et al. [52] found that technical training can help farmers to learn how to use new techniques, make farmers have more confidence in adopting green control techniques, and improve their intention to use them. Liu et al. [53] believed that when farmers’ efforts to use this technology are low, they will have more intention to adopt it. In this study, when farmers feel that the techniques are easier to use, they could be more inclined to continue to use the techniques.

However, risk perception might play a mediating role in the effect of effort expectancy on behavioral intention. If farmers believe that tea plants are greatly susceptible to pests and diseases, they would take strong measures to control them. That is, they would apply more chemical pesticides to protect tea plants from pests and diseases rather than adopting green control techniques. Although tea farmers perceive that green control techniques are easy to use, they also believe that green control techniques are not very effective in controlling pests and diseases due to their strong risk perception. Therefore, two hypotheses are proposed as follows:

**Hypothesis** **4** **(H4).**
*Effort expectancy has a positive effect on the behavioral intention of green control techniques.*


**Hypothesis** **4M** **(H4M).**
*Risk Perception of pests and diseases mediates the effect of effort expectancy on the behavioral intention of green control techniques.*


### 2.6. Social Influence

Social influence is the influence of important people around you or people who have a great influence on you to use techniques. Social influence has been confirmed that it has a significant effect on the behavioral intention of technique use [54,55,56], and it is an important driver for technique adoption [57]. Zeng et al. [58] stated that the popularization of green techniques is the formation of social consensus. Wu et al. [59] found that farmers who are more affected by the behavior of the agricultural techniques extension department and neighbors have a greater intention to adopt green techniques. Moreover, the long-term interaction between farmers enhances the sense of identity in techniques. They are more vulnerable to the influence of their neighbors’ behavior because of the existence of connecting power. Tzeling et al. [60] indicated that the frequency of communicating with neighbors would significantly affect their behavior and decision making. Liu et al. [53] indicated that neighbors have an impact on farmers’ intention to adopt green control techniques. That is, farmers, government, agricultural technicians, rural neighborhoods, and so on will have an impact on farmers’ adoption intention. Therefore, a hypothesis is posed as follows:

**Hypothesis** **5** **(H5).**
*Social influence has a positive effect on the behavioral intention of green control techniques.*


### 2.7. Facilitating Conditions

Facilitating conditions refer to a system that individuals think has certain technical and organizational conditions to facilitate the use of the system [22]. In the UTAUT framework, it has a smaller impact on behavioral intention [22], but it has an impact on the actual use of the system or techniques [48,61]. When the government provides equipment and technical support, farmers will be inclined to use the techniques. The agricultural subsidy policies implemented by the government can greatly influence the green production behavior of farmers [62,63]. That is, green control techniques may be widely implemented in tea planting if there are available facilities and resources to support farmers’ use. Thus, this study puts forward a hypothesis as follows:

**Hypothesis** **6** **(H6).**
*Facilitating conditions have a positive effect on farmers’ behavioral intention of green control techniques.*


## 3. Methodology

### 3.1. Questionnaire Design

The questionnaire consists of three parts: the first part is the basic information on demographic characteristics, and their planting and technology adoption behavior will be investigated. The second part is the core of the questionnaire, and questions are set under each of the 7 items. The third part mainly enquires about the tea harvest and income as well as the feeling of using green control techniques. The investigation design of four constructions (performance expectancy, effort expectancy, social influence, facilitating conditions) of the UTAUT is designed by referring to the research of previous literature [22,64,65]. Each item is designed with about 4 to 5 questions, using seven variables of a 5-point Likert scale (1 = strongly agree, 5 = strongly disagree). Due to commonly used in behavioral research, the Likert scale is chosen to organize the perception and subjective evaluation of technology adoption of respondents, and the structural equation model (SEM) is used in further statistical analysis. Appendix A presents the details of the questionnaire.

### 3.2. Data Collection

During the period between May and October 2019, this paper used stratified random sampling to randomly select nine prefecture-level cities from the main tea-producing regions in Sichuan, including Chengdu, Yaan, Leshan, Mianyang, Yibin, Bazhong, Meishan, Leshan, and Guangyuan. Then, two or three counties with a high concentration of business entities are selected in each city, and one or two townships are chosen in each county. The survey initially targeted two groups of tea farmers: green control techniques adopters and non-adopters, and the two-stage stratified random sampling procedure was applied. In the first step, using the information offered by the agricultural departments of the townships, the villages in each sample township were divided into two categories: green prevention and control demonstration villages and non-demonstration villages. Based on the principle of stratified random sampling, we selected one or two green prevention and control demonstration villages in each sample township for sampling. To ensure strong comparability between the two types of villages, we also considered the economic conditions, transportation conditions, and tea production potential of each village. In the second step, we obtained the list of all tea growers in each sample village with the help of the village committee. We randomly selected ten tea farmers from the demonstration villages of green prevention and control and ten tea farmers from the non-demonstration villages. Professionally trained investigators interviewed tea farmers one-on-one and recorded questionnaire information. Finally, we received 800 questionnaires in total, and 747 valid questionnaires remained after removing invalid questionnaires.

### 3.3. Data Analysis

The Smart-PLS 3.0 application is used to model a structural equation in this study. PLS-SEM (partial lead squares structural equation modeling) offered a structural equation modeling method, which integrates the advantages of principal component analysis, linear regression analysis, and typical correlation analysis. It is suitable for small data and large correlations. It has a higher statistical function, which is more conducive to the exploration and development of theoretical models. In addition, it takes into account the sample size, measurement range, and residual distribution. We divide it into two parts: the first part measures the validity and reliability of the research model and determines the fit. The second part is to test the structural relationship between latent variables to verify our hypotheses.

## 4. Results

### 4.1. Sample Characteristics

A total of 747 valid samples (Table 1) were collected from people from all over China, including 442 males (59.2%) and 691 middle-aged and elderly people over 49 (65.7%). Of these, 444 (59.5%) respondents have less than nine years of education, which accounts for more than half of the sample size. Most of the respondents were not cadres (92.6%). According to the results of mean and variance (see Table 2), respondents tend to associate high effort expectations with tea planting (M = 3.704, SD = 0.628) because their mean value is the highest and standard deviation is the lowest. There is little difference between the mean values of risk perception (M = 2.772, SD = 0.863), performance expectancy (M = 3.214, d = 0.880), and social influence (M = 2.301, SD = 0.705), and the standard deviation is relatively small. The mean value and standard deviation of intention are relatively small, indicating that respondents’ intention to use green control techniques is not so strong (M = 2.183, SD = 0.766).

### 4.2. Reliability and Validity

Table 3 presents the results of descriptive statistics. Cronbach’s Alpha (CA) is a good parameter to measure the model’s internal consistency. When the CA value is greater than 0.7, it is proved to be reliable [66]. According to the results, the CA values of all constructs are greater than 0.7, confirming that the model is consistent and reliable. The Composite Reliability (CR) value can also be used to measure the reliability of constructs, which is considered to be more advisable for measuring reliability than CA while using the PLS-SEM model [67]. In Table 3, all CR values are over 0.8, which meets the critical value standard of 0.7, indicating good reliability [68]. In addition, convergence validity can be used to test the degree of aggregation or correlation among indicators of the same construct, and greater than 0.5 indicates convergence validity. All AVE values except risk perception (0.582) in the results are bigger than 0.7, suggesting the results have a good convergence validity. The Fornell–Larcker criterion is often applied to judge discrimination reliability [69]. The bold part of the diagonal is the root sign value of AVE, which is required to be greater than the correlation coefficient with other constructs. According to these results, all of them conform to this rule, showing there is no evidence for the lack of discrimination validity.

Moreover, we obtain the autocorrelation result between each construct by checking the multicollinearity of them [67]. Variance Inflation Factor (VIF) is an indicator to measure the degree of multicollinearity in the multiple linear regression model. The VIF value greater than 5 indicates strong multicollinearity. Therefore, we eliminated the questionnaire items that do not meet the VIF standard. Finally, the coefficient of VIF of all variables is greater than 0.2 and less than 5, and the multicollinearity problem can be avoided.

### 4.3. Path Analysis

Path analysis was carried out by the SEM, and the results of path analysis are reported in Table 4 and Figure 2. First, the coefficient value of behavioral intention (H1) is 0.130 (*p* < 0.000), suggesting that behavioral intention has a positive effect on usage behavior. Its f^2^ value is 0.029, meaning that the effect is small. Second, risk perception (H2) is 0.118 (*p* < 0.000), which also confirms that risk perception harms behavioral intention. Its f^2^ value is 0.025, indicating that it has a small influence on behavioral intention. Third, the coefficient value of performance expectancy (H3) is 0.344 (*p* < 0.000), which means it can positively affect behavioral intention. The f^2^ value of performance expectancy is 0.128, indicating that performance expectancy has a moderate impact on behavioral intention. Fourth, the coefficient value of effort expectancy (H4) is 0.305 (*p* < 0.000) with an f^2^ value of 0.09, implying that effort expectancy positively affects behavioral intention, but the impact is weaker. Lastly, social influence (H5) has a small impact on behavioral intention because the coefficient value of 0.107 with an f^2^ value of 0.015 is the lowest. The R^2^ of behavioral intention suggests that risk perception, performance expectancy, effort expectancy, and social influence explain the 45.3% effect of green control techniques adoption intention.

Concerning usage behavior, we found that facilitating conditions have a significant positive influence on usage behavior (H6). The coefficient is the highest at 0.647 (*p* < 0.000), and its f^2^ value is also the highest at 0.712, suggesting a strong influence on usage behavior. In addition, the coefficient value of behavioral intention to usage behavior is 0.130 (*p* < 0.000), meaning that it can positively impact usage behavior but the impact is weak because of the small f^2^ value. The standardized regression coefficients indicate that both facilitating conditions and behavioral intention could positively impact the usage behavior significantly, but the effect of facilitating conditions is higher than expected.

### 4.4. Mediating Effect

In the mediating model, we put forward two hypotheses and show their indirect effect coefficients and confidence intervals. Bootstrap (*n* = 5000) is used to test the mediating effect of effort expectancy mediating risk perception on behavioral intention and effort expectancy mediating risk perception on behavioral intention. The standard of the test is to see whether the confidence interval contains 0. If it does, the original hypothesis should be rejected. Results indicate that significant moderations were triggered by performance effort on the paths between risk perception and behavioral intention. Risk perception also mediates the relationship between effort expectancy and behavioral intention. As shown in Table 5 and Figure 2, risk perception (H3M) has a significant indirect effect on behavioral intention (95% CI −0.237 to −0.078), and the hypothesis is accepted. Additionally, performance expectancy negatively moderates the interaction between risk perception and behavioral intention with an f^2^ value of 0.023. H4M also confirmed that the indirect effect of effort expectancy on behavioral intention can be mediated by risk perception, which is also statistically significant (95% CI −0.222 to −0.062). Both hypotheses of the mediation effect are proved.

## 5. Discussion

In China, pesticide overuse has become a major source of environmental pollution. As the main users of pesticides, farmers’ pesticide use behavior is directly related to environmental pollution. Generally speaking, most farmers are risk-averse and have little knowledge about the safe use of pesticides, which encourages farmers to use pesticides in excess [6]. Data from the National Bureau of Statistics show that before 2015, pesticide use in China was on a trend of increasing year by year. In order to curb the excessive use of chemical pesticides and ensure the quality and safety of agricultural products and ecological sustainability, the government introduced the *Zero Growth Action Plan for Pesticide Use by 2020* in 2015, which clearly stated that it was necessary to accelerate the transformation of pest prevention and control methods and vigorously promote green control techniques. Although the policy of pesticide use reduction has generated significant benefits in the past five years, farmers have not been very active in adopting green control techniques. This may be because green control techniques are a new alternative technique to green agriculture, requiring farmers to have the relevant knowledge and operational skills. However, the diffusion of green control techniques is difficult. In the case of the tea industry, most tea production areas are in the mountains. Due to the closed information, farmers cannot keep informed of the outside world’s environmental protection concepts and new agricultural technologies. At the same time, most farmers have a low level of education and have great difficulties in learning new knowledge and mastering new techniques. Tea is one of the special agricultural products in China, with a high economic value. Tea plants are vulnerable to pests and diseases, causing huge production losses. However, excessive use of pesticides can lead to excessive pesticide residues in tea, which brings disadvantages to farmers’ income and ecological environment. Therefore, it is important to study the intention and behavior of tea farmers to adopt green control techniques for the development of the tea industry and green agriculture.

This paper examines Chinese tea farmers’ intention and behavior to use green control techniques in the extended UTAUT framework and obtains some useful findings. First, performance expectancy, effort expectancy, and social influence have a positive effect on farmers’ behavioral intention to use green control technologies, which is consistent with the prediction of the classical UTAUT [10,19,42,43,44,70]. That is, the usefulness of techniques, perceived ease, and the influence of surrounding people will affect their adoption intention. In addition, risk perception of pests and diseases can reduce farmers’ behavioral intention to adopt green control techniques. Most farmers are risk-averse, and if they perceive a high pest threat, they would take strong measures to control pests and diseases (e.g., applying large amounts of pesticides) rather than adopting new green control techniques with uncertainty [34,36]. In addition, facilitating conditions can promote farmers to use green control techniques, which is consistent with the findings of Venkatesh et al. [22], Sykes et al. [61], and Nistor et al. [48]. Facilitating conditions including infrastructure, technical support, and financial subsidies can encourage farmers to use green control techniques [62,63]. Furthermore, this paper also tests two mediating mechanisms. On the one hand, the effect of risk perception on behavioral intention is mediated by performance expectancy. When the perceived risk is high (pests and diseases seriously threaten the quality and yield of tea), farmers tend to perceive green control techniques as being of little use [46,47]. Therefore, technicians need to focus on improving the performance of green control techniques to effectively prevent pests and diseases while protecting the environment. On the other hand, the effect of effort expectancy on behavioral intentions is mediated by risk perception. Easier use of the technique can lead tea farmers to perceive it as not effective enough for control, thus reducing their intention to adopt it.

Therefore, the government should strengthen policy incentives to promote the effective implementation of green control techniques. For example, appropriate subsidies and support should be given to farmers who use green control techniques, thereby enhancing their adoption intention of green agricultural techniques [71,72]. In addition, they should also pay attention to the introduction and promotion of green control techniques, expanding farmers’ scope of adopting new techniques, and improving farmers’ acceptance and recognition. Moreover, technology training should be conducted to reduce the usage difficulty of green control techniques. The most effective way to promote agricultural techniques is to carry out field schools [73]. Furthermore, the social learning function and information channel function of the social network is very important for adoption behavior [74]. Thus, it is also necessary to establish a good social learning network in the local area.

## 6. Conclusions

This paper explores the intention and behavior of tea farmers to use green control techniques in the UTAUT framework. The results showed that risk perception of pests and diseases can negatively impact the behavioral intention of green control techniques. Performance effect, effort expectancy, and social influence can enhance behavioral intention significantly, and facilitating conditions has a positive on usage behavior. In addition, performance expectancy plays a mediating role between risk perception and behavioral intention, while risk perception mediates the effect of effort expectancy on behavioral intention. Finally, some policy implications for promoting the diffusion and application of green control techniques are presented in the discussion section based on these findings.

Of course, the findings of our study may not be exactly generalizable, but they could at least provide some policy insights for promoting green agriculture in Asian countries with predominantly small-scale agriculture. Moreover, we have to acknowledge that this study has some limitations. First, this study is one of the few contributing to research on green control techniques adoption in tea planting. Some studies about green control techniques added factors such as environmental policy, environmental tax, carbon emissions, supply chain, and so on, and this paper mainly considers the risk perception from farmers’ perspectives. Therefore, we can also consider new factors to study farmers’ micro behavior in the future. Second, some of the population characteristics in this study are relatively dense. For example, most of the samples are middle-aged people and people who are not cadres, so future research can be more decentralized in selecting samples. Third, due to the complexity of China’s agricultural management system, the types of farmers are mainly small farmers, family farms, and agricultural cooperatives. Their intention toward the adoption of green control techniques would be different. As a result, different types of farmers could be studied in the future.

## Figures and Tables

**Figure 1 ijerph-19-08465-f001:**
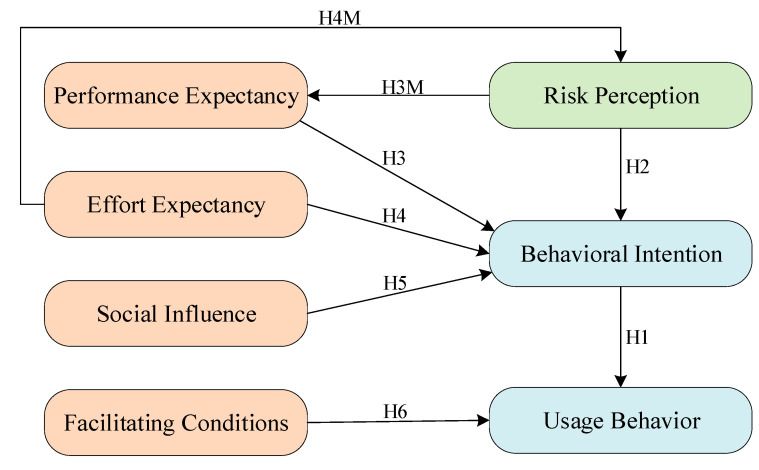
Conceptual framework.

**Figure 2 ijerph-19-08465-f002:**
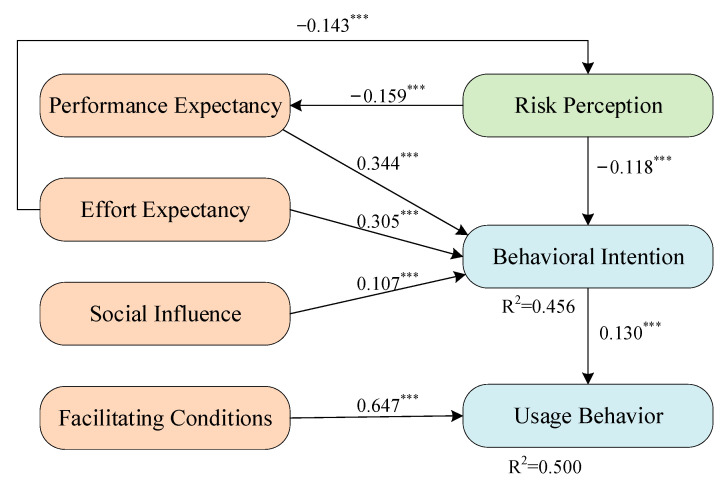
Path coefficients. Note: *** represents *p* < 0.01.

**Table 1 ijerph-19-08465-t001:** Sample characteristics.

Variables	Definitions	Frequency	Proportion
Gender	Female	442	59.2%
	Male	305	40.8%
Age	Between 18 and 28 years old	5	0.7%
	Between 29 and 38 years old	51	6.8%
	Between 39 and 48 years old	199	26.8%
	Between 49 and 58 years old	261	34.9%
	59 years old and above	231	30.8%
Education	3 schooling years and below	71	9.5%
	Between 3 and 6 schooling years	161	21.6%
	Between 6 and 9 schooling years	283	37.9%
	9 schooling years and above	232	31.0%
Cadre	Cadre	55	7.4%
	Non-cadre	692	92.6%

**Table 2 ijerph-19-08465-t002:** Descriptive statistics.

Variables	No. Items	Mean	SD
Risk Perception	4	2.772	0.863
Performance Expectancy	5	3.214	0.880
Effort Expectancy	5	3.704	0.628
Social Influence	4	2.301	0.705
Facilitating Conditions	5	2.576	0.797
Behavior Intention	3	2.183	0.766
Usage Behavior	4	3.200	1.068

Note: SD represents the standard deviation.

**Table 3 ijerph-19-08465-t003:** Construct correlation matrix and reliability measurement.

	CA	CR	AVE	VIF	UB	BI	RP	PE	EE	SI	FC
UB	0.936	0.954	0.839	-	**0.916**						
BI	0.902	0.939	0.837	1.172	0.378	**0.915**					
RP	0.733	0.835	0.582	1.031	0.109	−0.214	**0.763**				
PE	0.916	0.927	0.750	1.704	0.404	0.599	−0.150	**0.866**			
EE	0.909	0.932	0.734	1.885	0.478	0.591	−0.135	0.627	**0.856**		
SI	0.917	0.941	0.800	1.393	0.597	0.412	−0.024	0.424	0.511	**0.894**	
FC	0.925	0.944	0.770	1.172	0.696	0.383	0.015	0.511	0.565	0.683	**0.878**

Note: UB: Usage Behavior; BI: Behavioral Intention; RP: Risk Perception; PE: Performance Expectancy; EE: Effort Expectancy; SI: Social Influence; FC: Facilitating Conditions; CA: Cronbach’s Alpha; CR: Composite Reliability; AVE: Average Variance Extracted; VIF: Variance Inflation Factors. The bolded values on the diagonal are the square root of AVE.

**Table 4 ijerph-19-08465-t004:** Path coefficient.

Hypotheses	Beta	*p*-Value	R^2^	f^2^	Decision
H1: BI-UB	0.130	0.000	0.500	0.029	Accept
H6: FC-UB	0.647	0.000		0.712	Accept
H2: RP-BI	−0.118	0.000	0.456	0.025	Accept
H3: PE-BI	0.344	0.000		0.128	Accept
H4: EE-BI	0.305	0.000		0.090	Accept
H5: SI-BI	0.107	0.000		0.015	Accept

Note: UB: Usage Behavior; BI: Behavioral Intention; RP: Risk Perception; PE: Performance Expectancy; EE: Effort Expectancy; SI: Social Influence; FC: Facilitating Conditions.

**Table 5 ijerph-19-08465-t005:** Results of mediation.

Hypotheses	Beta	LLCI	ALSO	Results
H3M: PE mediates the relationship between RP and BI	−0.159	−0.237	−0.078	Mediation
H4M: RP mediates the relationship between EE and BI	−0.143	−0.222	−0.062	Mediation

Note: PE: Performance Expectancy; RP: Risk Perception; BI: Behavioral Intention; EE: Effort Expectancy.

## Data Availability

The data presented in this study are available within the article.

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
