# Peer review of "Effects of Risk Perception of Pests and Diseases on Tea Famers’ Green Control Techniques Adoption"

_ijerph, 2022, doi:10.3390/ijerph19148465_

Round 1
Reviewer 1 Report
Recently, the study on the adoption of green production technology and its influencing factors has become a hot topic in the field of resources and environment. The author's topic selection has certain significance, and the empirical study is quite standard. However, there are still outstanding problems in variable measurement, marginal contribution and condensing of key scientific issues in the whole study. Specific opinions are as follows:
(1) The marginal contribution of research is not strong. In the field of social psychology, few studies have looked at the adoption of green production technologies, the authors say. However, as far as I know, no matter in China or in the world, there are a lot of researches on farmers' adoption of green production technology using UTAUT model. Therefore, in my opinion, the author's marginal contribution is not clear or the author's claim of innovation simply does not hold water.
(2) Research lacks clarity on key scientific issues. Farmers' adoption of green production technology is a good topic range. However, for tea farmers, what are the prominent problems in their adoption of green production technology? Is this problem caused by risk perception? Due to the author's lack of awareness of the problem, the whole study just uses fancy models to verify a common sense.
(3) The measurement of variables lacks the corresponding basis, and the corresponding reliability and validity test is not done. The core keyword mentioned in the title of the author is "Risk Perception of Pest Disease", but the relevant entries in Table 1 of Annex A are not related to the Perception of Pest and Disease, so the overall research design of the author is problematic and irrelevant.
(4) The study lacks in-depth discussion. Although there is a discussion part in current studies, there is a lack of in-depth discussion and comparative analysis of similar studies.
Author Response
Comments from Reviewer 1
Recently, the study on the adoption of green production technology and its influencing factors has become a hot topic in the field of resources and environment. The author's topic selection has certain significance, and the empirical study is quite standard. However, there are still outstanding problems in variable measurement, marginal contribution and condensing of key scientific issues in the whole study. Specific opinions are as follows:
- The marginal contribution of research is not strong. In the field of social psychology, few studies have looked at the adoption of green production technologies, the authors say. However, as far as I know, no matter in China or in the world, there are a lot of researches on farmers' adoption of green production technology using UTAUT model. Therefore, in my opinion, the author's marginal contribution is not clear or the author's claim of innovation simply does not hold water.
- Research lacks clarity on key scientific issues. Farmers' adoption of green production technology is a good topic range. However, for tea farmers, what are the prominent problems in their adoption of green production technology? Is this problem caused by risk perception? Due to the author's lack of awareness of the problem, the whole study just uses fancy models to verify a common sense.
Response: Thanks for raising this suggestion to help in improving the quality of our manuscript. On the one hand, the search process of domestic and international databases revealed that there were not many studies using UTAUT to study farmers' green agricultural techniques usage behavior. On the other hand, the key scientific issues of this paper and the prominent problems in the adoption of green control techniques are stated in the introduction section. The revised introduction is as follows:
"1. Introduction
China is a major tea-producing country and tea has a long history in China [1-2]. As an important cash crop, tea is the basis for many rural farmers to sustain their livelihoods. Therefore, the development of the tea industry is crucial for tea farmers. However, tea plants are susceptible to a variety of pests such as nematodes, termites, cockchafer grub, stem pests, stem borer, leaf pests, and all mite species. If these pests and diseases are not effectively controlled, they could cause 10% to 20% of yield losses [3-4]. Since tea farmers are risk-averse, they have to apply pesticides to rapidly control pests and diseases for reducing production losses [5-6]. However, excessive usage of pesticides for reducing losses has created a serious problem of agricultural pollution. Some of the pesticide residues enter the ecosystem through the atmosphere, soil, and groundwater, causing incalculable damage to the ecosystem [7], and the pesticide residues on tea leaves can affect human health when brewed [4].
To curb the excessive use of chemical pesticides and ensure food quality and safety and ecological sustainability, the Chinese government emphasizes the promotion and application of green control techniques. Similar to integrated pest management (IPM) [8], green control techniques state effective control of crop pests and diseases by reducing chemical pesticide use and adopting techniques such as agricultural control, physical control, biological control, ecological regulation, and the scientific pesticide use to ensure the agricultural production safety, quality safety, and agroecological environment safety, as well as to improve farmers' welfare. However, data from the Ministry of Agriculture and Rural Development showed that by the end of 2021, the adoption rate of green control techniques in China still had not reached 50%. The reasons for the low adoption rate can be attributed to several aspects. First, although the adoption of green control techniques can bring positive environmental effects, their effectiveness in controlling pests and diseases is not immediate compared to the traditional means of applying chemical pesticides [6]. Second, the adoption of green control techniques requires farmers to learn the knowledge and operational skills, and even to shoulder the high cost of technology transfer and usage risk [9]. Finally, farmers have insufficient knowledge about pesticide application and weak awareness of ecological environment protection [5].
Moreover, many studies have shown that risk perception is one of the main factors influencing farmers' adoption of agricultural techniques. For example, Liu and Huang [10] found that risk-averse farmers tended to apply more pesticides. Fernandez-Cornejo et al. [11] pointed out that farmers who adopted the IPM tended to be less risk-averse compared to non-adopters. Li et al. [12] found that risk perception positively mediated the relationship between farmer differences and soil&water conservation techniques. And Pingali et al. [13] argued that the greater the risk farmers perceived, the more they intend to adopt pest control. The high risk of pests and diseases could lead to crop yield reduction, thus affecting their choice of control techniques. To our knowledge, although studies have discussed the impact of risk perception on pesticide use and sustainable technique adoption in agriculture, there is little literature examining the impact of risk perception of pests and diseases on the usage behavior and behavioral intentions of green control techniques. In recent years, many countries have emphasized sustainable and green development in agriculture. The development of green agriculture implies the transformation of agricultural production methods, and the transformation of control methods is also a key to it. It is well-known that the promotion and application of green control techniques have a positive effect on the improvement of the ecological environment and agricultural product quality. Therefore, it is important to explore the influence of risk perception of pests and diseases on the adoption of green control techniques by tea farmers for the development of policies related to the promotion of green agricultural techniques. As alluded above, this paper involves the risk perception of pests and diseases in the Unified Theory of Acceptance and Use of Technology (UTAUT) framework, using the micro-data of tea farmers from Sichuan province, China, attempting to explore the impact and mechanism of risk perception on tea farmers' adoption of green control techniques. We found that risk perception of pests and diseases has a negative effect on the behavioral intention of green control techniques. Additionally, performance expectancy, effort expectancy, and social influence have positive effects on behavioral intention. And facilitating conditions could positively affect usage behavior. Furthermore, performance expectancy can mediate the influence of risk perception on behavioral intention, and risk perception can mediate the effect of effort expectancy on behavioral intention.
This paper has some contributions. First, although the UTAUT has been used to study farmers' adoption of electronic commerce, communication technologies, Internet of Things technologies, and project payments [14-20], they have been rarely applied in the context of green agricultural techniques. In this paper, we introduce risk perception of pests and diseases into the original UTAUT framework in an attempt to explore the effect of risk perception on farmers' behavioral intention and usage behavior of green control techniques and contribute to the existing literature. Second, the data used in this paper is more representative because Sichuan is a major tea-producing province [4] and sufficient samples can also improve the estimation results under the structural equation model (SEM). Third, we also identified mediating effects of risk perception and performance expectancy. On the one hand, risk perception plays a mediating role between effort expectancy and behavioral intention. On the other hand, performance expectancy also mediates the relationship between risk perception and behavioral intention. These results are new findings for relevant studies and help us to understand the impact of risk perception of pests and diseases on the adoption of green control techniques at a deeper level. Fourth, our findings can provide policy implications for the government in making policies for the promotion of green control techniques as well as green agriculture development.
The remainders of this paper are shown as follows: Section 2 explains the theory and hypotheses; Section 3 introduces the questionnaire and data source; Sections 4 and 5 report the results and make a discussion, respectively; Section 6 states the conclusions, limitations, and future research."
- The measurement of variables lacks the corresponding basis, and the corresponding reliability and validity test is not done. The core keyword mentioned in the title of the author is "Risk Perception of Pest Disease", but the relevant entries in Table 1 of Annex A are not related to the Perception of Pest and Disease, so the overall research design of the author is problematic and irrelevant.
Response: Thank you for your comment. According to the measurement scales used in existing studies (Venkatesh et al., 2003; Duyck et al., 2008; Venkatesh and Zhang, 2010) and combining the Chinese reality, we designed this questionnaire. Meanwhile, we tested the reliability and validity, and the results showed that the questionnaire and data sets were valid.
- The study lacks in-depth discussion. Although there is a discussion part in current studies, there is a lack of in-depth discussion and comparative analysis of similar studies
Response: We are grateful for your advice. The revised discussion is as follows:
"5. Discussion
In China, pesticide overuse has become a major source of environmental pollution. As the main users of pesticides, farmers' pesticide use behavior is directly related to environmental pollution. Generally speaking, most farmers are risk-averse and have little knowledge about the safe use of pesticides, which encourages farmers to use pesticides in excess [6]. Data from the National Bureau of Statistics shows that before 2015, pesticide use in China was on a trend of the increasing year by year. In order to curb the excessive use of chemical pesticides and ensure the quality and safety of agricultural products and ecological sustainability, the government introduced the Zero Growth Action Plan for Pesticide Use by 2020 in 2015, which clearly stated that it was necessary to accelerate the transformation of pest prevention and control methods and vigorously promote green control techniques. Although the policy of pesticide use reduction has generated significant benefits in the past five years, farmers have not been very active in adopting green control techniques. This may be because green control techniques are a new alternative technique to green agriculture, requiring farmers to have the relevant knowledge and operational skills. However, the diffusion of green control techniques is difficult. In the case of the tea industry, most tea production areas are in the mountains. Due to the closed information, farmers cannot keep informed of the outside world's environmental protection concepts and new agricultural technologies. At the same time, most farmers have a low level of education and have great difficulties in learning new knowledge and mastering new techniques. Tea is one of the special agricultural products in China, with a high economic value. Tea plants are vulnerable to pests and diseases, causing huge production losses. However, excessive use of pesticides can lead to excessive pesticide residues in tea, which brings disadvantages to farmers' income and ecological environment. Therefore, it is important to study the intention and behavior of tea farmers to adopt green control techniques for the development of the tea industry and green agriculture.
This paper examines Chinese tea farmers' intention and behavior to use green control techniques in the extended UTAUT framework and obtains some useful findings. First, performance expectancy, effort expectancy, and social influence have a positive effect on farmers' behavioral intention to use green control technologies, which is consistent with the prediction of the classical UTAUT [10, 19, 42-44, 70]. That is, the usefulness of techniques, perceived ease, and the influence of surrounding people will affect their adoption intention. In addition, Risk perception of pests and diseases can reduce farmers' behavioral intention to adopt green control techniques. Most farmers are risk-averse, and if they perceive a high pest threat, they would take strong measures to control pests and diseases (e.g., applying large amounts of pesticides) rather than adopting new green control techniques with uncertainty [34, 36]. Besides, facilitating conditions can promote farmers to use green control techniques, which is consistent with the findings of Venkatesh et al. [22], Sykes et al. [61], and Nistor et al. [48]. Facilitating conditions including infrastructure, technical support, and financial subsidies can encourage farmers to use green control techniques [62-63]. Furthermore, this paper also tests two mediating mechanisms. On the one hand, the effect of risk perception on behavioral intention is mediated by performance expectancy. When the perceived risk is high (pests and diseases seriously threaten the quality and yield of tea), farmers tend to perceive green control techniques as being of little use [46-47]. Therefore, technicians need to focus on improving the performance of green control techniques to effectively prevent pests and diseases while protecting the environment. On the other hand, the effect of effort expectancy on behavioral intentions is mediated by risk perception. Easier use of the technique can lead tea farmers to perceive it as not effective enough for control, thus reducing their intention to adopt it.
Therefore, the government should strengthen policy incentives to promote the effective implementation of green control techniques. For example, appropriate subsidies and support should be given to farmers who use green control techniques, thereby enhancing their adoption intention of green agricultural techniques [71-72]. In addition, it should also pay attention to the introduction and promotion of green control techniques, expanding farmers' scope of adopting new techniques, and improving farmers' acceptance and recognition. Moreover, technology training should be conducted to reduce the usage difficulty of green control techniques. The most effective way to promote agricultural techniques is to carry out field schools [73]. Furthermore, the social learning function and information channel function of the social network is very important for adoption behavior [74]. Thus, it is also necessary to establish a good social learning network in the local area."
Reviewer 2 Report
The manuscript contains interesting research results. The manuscript is well developed, the test results come from a reliable sample, and the statistical analysis of the data allowed for correct conclusions. However, after reading the manuscript, I have some doubts as to the location of this data in an international journal. While the role of China in the global distribution of dried tea is not in doubt, the global consumer will be more interested in the results of research on the actual pesticide residue (in the context of choosing safe food) than the surveyed belief / perception of Chinese farmers about green control. In order to achieve the desired (perhaps) effect of popularizing green control in the cultivation of tea, maybe it is better to publish this data in a journal on a regional or national scale? Why not find a way to expose the meaning of the results to the global reader? I leave the decision to the MDPI Editor.
Author Response
Comments from Reviewer 2
The manuscript contains interesting research results. The manuscript is well developed, the test results come from a reliable sample, and the statistical analysis of the data allowed for correct conclusions. However, after reading the manuscript, I have some doubts as to the location of this data in an international journal. While the role of China in the global distribution of dried tea is not in doubt, the global consumer will be more interested in the results of research on the actual pesticide residue (in the context of choosing safe food) than the surveyed belief / perception of Chinese farmers about green control. In order to achieve the desired (perhaps) effect of popularizing green control in the cultivation of tea, maybe it is better to publish this data in a journal on a regional or national scale? Why not find a way to expose the meaning of the results to the global reader? I leave the decision to the MDPI Editor.
Response: Thank you for your comment. Although we used Chinese tea farmers as the study sample to explore farmers' intention and behavior to adopt green control techniques, our findings can provide at least some policy implications for green agricultural development in many developing countries in Asia where small-scale agriculture dominates. Additionally, many papers published in MDPI were sampled from a certain small region.
Guo, H., Zhao, W., Pan, C., Qiu, G., Xu, S., & Liu, S. (2022). Study on the Influencing Factors of Farmers’ Adoption of Conservation Tillage Technology in Black Soil Region in China: A Logistic-ISM Model Approach. International Journal of Environmental Research and Public Health, 19(13), 7762.
Khan, N., Ray, R. L., Kassem, H. S., & Zhang, S. (2022). Mobile Internet Technology Adoption for Sustainable Agriculture: Evidence from Wheat Farmers. Applied Sciences, 12(10), 4902.
Kassem, H.S.; Hussein, M.A.; Ismail, H. The Dilemma of Fraudulent Pesticides in the Agrifood Sector: Analysis of Factors Affecting Farmers’ Purchasing Behavior in Egypt. Agronomy 2022, 12, 1626.
Si R, Yao Y, Zhang X, Lu Q, Aziz N. Exploring the Role of Contiguous Farmland Cultivation and Adoption of No-Tillage Technology in Improving Transferees’ Income Structure: Evidence from China. Land. 2022; 11(4):570.
Reviewer 3 Report
the comment is attached below

Author Response
Comments from Reviewer 3
The article examines the factors influencing the adoption of green control techniques in tea cultivation by Chinese farmers. Among the factors, the psychological and mental characteristics of farmers were taken into account. The authors have documented the research gap on this topic. The challenges facing modern agriculture are related to the production of healthy, safe food and the fight against climate change. From this point of view, the choice of the topic should be considered correct and important. The selection of methods is correct, the interpretation of the results is correct. The work makes a good impression, but has some weaknesses:
- The purpose of the work must be clearly emphasized. In this case, there is information
about what the research brings new to learning (in the "Introduction" section), but no specific
purpose for the work.
- There are mental shortcuts, eg "This paper explains the theoretical model and empirical ..."
(line 75) - what is this model about?
Response: Thanks for raising this suggestion to help in improving the quality of our manuscript. The revised introduction is as follows:
"1. Introduction
China is a major tea-producing country and tea has a long history in China [1-2]. As an important cash crop, tea is the basis for many rural farmers to sustain their livelihoods. Therefore, the development of the tea industry is crucial for tea farmers. However, tea plants are susceptible to a variety of pests such as nematodes, termites, cockchafer grub, stem pests, stem borer, leaf pests, and all mite species. If these pests and diseases are not effectively controlled, they could cause 10% to 20% of yield losses [3-4]. Since tea farmers are risk-averse, they have to apply pesticides to rapidly control pests and diseases for reducing production losses [5-6]. However, excessive usage of pesticides for reducing losses has created a serious problem of agricultural pollution. Some of the pesticide residues enter the ecosystem through the atmosphere, soil, and groundwater, causing incalculable damage to the ecosystem [7], and the pesticide residues on tea leaves can affect human health when brewed [4].
To curb the excessive use of chemical pesticides and ensure food quality and safety and ecological sustainability, the Chinese government emphasizes the promotion and application of green control techniques. Similar to integrated pest management (IPM) [8], green control techniques state effective control of crop pests and diseases by reducing chemical pesticide use and adopting techniques such as agricultural control, physical control, biological control, ecological regulation, and the scientific pesticide use to ensure the agricultural production safety, quality safety, and agroecological environment safety, as well as to improve farmers' welfare. However, data from the Ministry of Agriculture and Rural Development showed that by the end of 2021, the adoption rate of green control techniques in China still had not reached 50%. The reasons for the low adoption rate can be attributed to several aspects. First, although the adoption of green control techniques can bring positive environmental effects, their effectiveness in controlling pests and diseases is not immediate compared to the traditional means of applying chemical pesticides [6]. Second, the adoption of green control techniques requires farmers to learn the knowledge and operational skills, and even to shoulder the high cost of technology transfer and usage risk [9]. Finally, farmers have insufficient knowledge about pesticide application and weak awareness of ecological environment protection [5].
Moreover, many studies have shown that risk perception is one of the main factors influencing farmers' adoption of agricultural techniques. For example, Liu and Huang [10] found that risk-averse farmers tended to apply more pesticides. Fernandez-Cornejo et al. [11] pointed out that farmers who adopted the IPM tended to be less risk-averse compared to non-adopters. Li et al. [12] found that risk perception positively mediated the relationship between farmer differences and soil&water conservation techniques. And Pingali et al. [13] argued that the greater the risk farmers perceived, the more they intend to adopt pest control. The high risk of pests and diseases could lead to crop yield reduction, thus affecting their choice of control techniques. To our knowledge, although studies have discussed the impact of risk perception on pesticide use and sustainable technique adoption in agriculture, there is little literature examining the impact of risk perception of pests and diseases on the usage behavior and behavioral intentions of green control techniques. In recent years, many countries have emphasized sustainable and green development in agriculture. The development of green agriculture implies the transformation of agricultural production methods, and the transformation of control methods is also a key to it. It is well-known that the promotion and application of green control techniques have a positive effect on the improvement of the ecological environment and agricultural product quality. Therefore, it is important to explore the influence of risk perception of pests and diseases on the adoption of green control techniques by tea farmers for the development of policies related to the promotion of green agricultural techniques. As alluded above, this paper involves the risk perception of pests and diseases in the Unified Theory of Acceptance and Use of Technology (UTAUT) framework, using the micro-data of tea farmers from Sichuan province, China, attempting to explore the impact and mechanism of risk perception on tea farmers' adoption of green control techniques. We found that risk perception of pests and diseases has a negative effect on the behavioral intention of green control techniques. Additionally, performance expectancy, effort expectancy, and social influence have positive effects on behavioral intention. And facilitating conditions could positively affect usage behavior. Furthermore, performance expectancy can mediate the influence of risk perception on behavioral intention, and risk perception can mediate the effect of effort expectancy on behavioral intention.
This paper has some contributions. First, although the UTAUT has been used to study farmers' adoption of electronic commerce, communication technologies, Internet of Things technologies, and project payments [14-20], they have been rarely applied in the context of green agricultural techniques. In this paper, we introduce risk perception of pests and diseases into the original UTAUT framework in an attempt to explore the effect of risk perception on farmers' behavioral intention and usage behavior of green control techniques and contribute to the existing literature. Second, the data used in this paper is more representative because Sichuan is a major tea-producing province [4] and sufficient samples can also improve the estimation results under the structural equation model (SEM). Third, we also identified mediating effects of risk perception and performance expectancy. On the one hand, risk perception plays a mediating role between effort expectancy and behavioral intention. On the other hand, performance expectancy also mediates the relationship between risk perception and behavioral intention. These results are new findings for relevant studies and help us to understand the impact of risk perception of pests and diseases on the adoption of green control techniques at a deeper level. Fourth, our findings can provide policy implications for the government in making policies for the promotion of green control techniques as well as green agriculture development.
The remainders of this paper are shown as follows: Section 2 explains the theory and hypotheses; Section 3 introduces the questionnaire and data source; Sections 4 and 5 report the results and make a discussion, respectively; Section 6 states the conclusions, limitations, and future research."
- Numbering of hypotheses is not understandable, there is chaos in numbering - first, there is hypothesis 6, then 2, 1M, etc. - why such numbering? All these issues need clarification
Response: We thank the reviewer for this comment. The number of hypotheses was modified as follows:
"2. Theory and hypotheses
2.1. Unified Theory of Acceptance and Use of Technology
The unified Theory of Acceptance and Use of Technology (UTAUT) is often used to determine the predictors affecting technology acceptance. This model has been proven to have good predictability [21]. There are many factors related to technology adoption that can affect usage behavior. It involves four key determinants: performance expectancy, effort expectancy, social influence, and facilitating conditions. Behavioral intention is determined by three factors: performance expectation, effect expectation, and social influence. And facilitating conditions affect usage behavior. The main model [22] is rooted in the traditional techniques acceptance model (TAM) [23]. Based on the conceptual and empirical similarity of the eight classical models [19], they are integrated into a single model, making the UTAUT an important model for studying the adoption of technology [24-25]. In agriculture, the UTAUT has been used to study farmers' intentions and behavior in using technologies such as Internet of Things technologies [17-18], electronic commerce [15, 20], communication technologies [14, 16], smartphone [26], and energy project payments [19]. However, to our knowledge, the UTAUT has rarely been applied in the field of green agricultural techniques, so this area is worth exploring. Therefore, this paper contributes to the literature in this area by using the UTAUT to study the intention and behavior of Chinese tea farmers in adopting green control techniques. More importantly, this paper introduces risk perception of pests and diseases based on the original UTAUT to analyze the effect of risk perception on the adoption of green control techniques.
2.2. Behavioral intention and usage behavior
Behavioral intention means the intention degree of using and applying new techniques, which is also defined as the probability for someone to make a conscious plan to achieve future certain behaviors [27]. Usage behavior refers to the actual degree of users' technique adoption [22]. In many behavioral theoretical models, behavioral intention is a good factor for behavioral prediction [23, 28-19]. The increasing studies confirmed that when people have a stronger intention to use technology, they are more likely to use it [17, 19, 22, 24]). Thus, this study puts forward a hypothesis as follows:
Hypothesis 1 (H1): Behavioral intention has a positive effect on farmers' usage behavior of green control techniques.
2.3. Risk Perception
Risk perception is defined as the perception of adverse consequences caused by an individual engaging in a specific activity (such as using a specific technique) under uncertain conditions [30-31]. It is regarded as an important factor in predicting human behavior. Some scholars introduced the variable of risk perception into the UTAUT framework and discussed the effect. For instance, Khatimah [32] added the variable of risk perception into the UTAUT model, indicating that risk perception would have an impact on people's behavioral intentions. Chen [33] studied the potential application of mobile payment and found that the risk perception of people is the direct reason why they have the intention to use mobile phones.
Pest disease is considered to be the one of biggest risks faced by farmers because it will have a decisive impact on their output and income. To control pests and diseases and reduce yield losses, they will apply pesticides. Pingali [34] studied the rice production behavior of farmers in the Philippines and found that farmers often apply excessive pesticides to avoid the risk of pests and diseases. Liu and Huang [35] also indicated that farmers with higher risk perception would choose to apply more pesticides in planting. However, such behavior would cause great damage to the environment. That is, in the face of greater risks, farmers' lower awareness of environmental protection may be one of the reasons for the low intention to adopt green control techniques. Additionally, the research of Chu and Li [36] showed that the degree of farmers' risk preference also affects their behavior in adopting green control techniques. Preferred risk farmers can accept the uncertainty of adopting new techniques and have more intention to apply them. Therefore, this study adds the variable of risk perception of pests and diseases into the UTAUT framework to understand farmers' techniques adoption behavior. The hypothesis is proposed as follows:
Hypothesis 2 (H2): Risk perception has a negative effect on the behavioral intention of green control techniques.
2.4. Performance Expectancy
Performance expectancy is the degree to which farmers believe that the use of green control technologies can help them obtain benefits [22]. Performance expectation is the one of strongest predictors of behavior intention of techniques adoption. It is important in predicting farmers' adoption of techniques [37-38]. Many studies confirmed that performance expectancy can positively influence people's technology adoption [39-44]. For example, Hans et al. [45] showed that if techniques can bring higher efficiency, the intention of farmers to adopt it will be stronger. Liu and Wang [10] found that the more useful farmers feel when using techniques, the stronger their behavioral intention is.
Moreover, farmers generally have limited knowledge of green control techniques, and they are uncertain about their performance. For instance, Martins et al. [46] indicated that when people perceive a high risk, they may think the convenience brought by Internet banking is not so strong, so their behavioral intention is low. Tai and Ku [47] believed that when stock investors trade stocks through mobile devices, they are required to provide some private information, which may be used by some hackers for improper operations. Therefore, when investors are aware of the risks, they may decide to give up the potential benefits of using mobile stock trading. Most of the current studies have shown that when people perceive a high risk of using such a system or technology, they feel that the benefits of the system will decline. In this study, if farmers perceive a higher risk of pests and diseases, they would not believe that green control techniques produce significant performance effects, thereby leading to a lower intention to use them. Therefore, this study presents two hypotheses as follows:
Hypothesis 3 (H3): Performance expectancy has a positive effect on behavioral intention to use green control techniques.
Hypothesis 3M (H3M): Performance expectancy of green control techniques mediates the effect of risk perception on the behavioral intention of green control techniques.
2.5. Effort Expectancy
Effort expectancy is the perceived ease associated with the use of techniques [22]. It comes from the TAM model and is adapted from "perceived ease of use". Some scholars found that effort expectancy positively impacts behavioral intention [43, 48-49]. Davis et al. [50] and Sorebo and Eikebrokk [51] found that farmers' intentions would be higher when they think that the difficulty of applying the techniques is small. Paul et al. [52] found that technical training can help farmers to learn how to use new techniques, make farmers have more confidence in adopting green control techniques, and improve their intention to use them. Liu et al. [53] believed that when farmers' efforts to use this technology are low, they will have more intention to adopt it. In this study, when farmers feel that the techniques are easier to use, they could be more inclined to continue to use the techniques.
However, risk perception might play a mediating role in the effect of effort expectancy on behavioral intention. If farmers believe that tea plants are greatly susceptible to pests and diseases, they would take strong measures to control them. That is, they would apply more chemical pesticides to protect tea plants from pests and diseases rather than adopting green control techniques. Although tea farmers perceive that green control techniques are easy to use, they also believe that green control techniques are not very effective in controlling pests and diseases due to their strong risk perception. Therefore, two hypotheses are proposed as follows:
Hypothesis 4 (H4): Effort expectancy has a positive effect on the behavioral intention of green control techniques.
Hypothesis 4M (H4M): Risk Perception of pests and diseases mediates the effect of effort expectancy on the behavioral intention of green control techniques.
2.6. Social Influence
Social influence is the influence of important people around you or people who have a great influence on you to use techniques. Social influence has been confirmed that it has a significant effect on the behavioral intention of technique use [54-56], and it is an important driver for technique adoption [57]. Zeng et al. [58] stated that the popularization of green techniques is the formation of social consensus. Wu et al. [59] found that farmers who are more affected by the behavior of the agricultural techniques extension department and neighbors have more intention to adopt green techniques. Moreover, the long-term interaction between farmers enhances the sense of identity in techniques. They are more vulnerable to the influence of their neighbors' behavior because of the existence of connecting power. Tzeling et al. [60] indicated that the frequency of communicating with neighbors would significantly affect their behavior and decision-making. Liu et al. [53] indicated that neighbors have an impact on farmers' intention to adopt green control techniques. That is, farmers, government, agricultural technicians, rural neighborhoods, and so on will have an impact on farmers' adoption intention. Therefore, a hypothesis is posed as follows:
Hypothesis 5 (H5): Social influence has a positive effect on the behavioral intention of green control techniques.
2.7. Facilitating Conditions
Facilitating conditions refer to a system that individuals think has certain technical and organizational conditions to facilitate the use of the system [22]. In the UTAUT framework, it has a smaller impact on behavioral intention [22], but it has an impact on the actual use of the system or techniques [48, 61]. When the government provides equipment and technical support, farmers will be inclined to use the techniques. The agricultural subsidy policies implemented by the government can greatly influence the green production behavior of farmers [62-63]. That is, green control techniques may be widely implemented in tea planting if there are available facilities and resources to support farmers' use. Thus, this study puts forward a hypothesis as follows:
Hypothesis 6 (H6): Facilitating conditions have a positive effect on farmers' behavioral intention of green control techniques.
Round 2
Reviewer 1 Report
I have no other comments.